# Mental health recovery for survivors of modern slavery: grounded theory study protocol

Nicola Wright,[1] Emina Hadziosmanovic ![ORCID],[1] Minh Dang,[2,3] Kevin Bales,[2] Caroline Brookes,[4] Melanie Jordan,[5] Mike Slade,[1] Lived Experience Research Advisory Board

¹School of Health Sciences, University of Nottingham, Nottingham, UK
²School of Politics and International Relations, University of Nottingham, Nottingham, UK
³Survivor Alliance, Birmingham, UK
⁴Emergency Preparedness Resilience & Response (EPRR), Nottinghamshire Healthcare NHS Foundation Trust, Nottingham, UK
⁵School of Sociology and Social Policy, University of Nottingham, Nottingham, UK

**Correspondence to**
Dr Nicola Wright;
nicola.wright@nottingham.ac.uk

## ABSTRACT

**Introduction** Slavery and human trafficking are crimes involving the violation of human rights and refer to exploitative situations where an individual cannot refuse or leave due to threats, coercion or abuse of power. Activities involving slavery include forced labour exploitation, forced sexual exploitation, forced marriage and servitude. Epidemiological studies show high levels of mental health need and poor provision of appropriate support for survivors. What mental health recovery means to victims/survivors and how it could be promoted is under-researched.

**Methods and analysis** A grounded theory study based on individual interviews will be undertaken. Survivors across the UK will be identified and recruited from non-governmental organisations and via social media. As per grounded theory methodology, data collection and analysis will be undertaken concurrently and recruitment will continue until theoretical saturation is reached. It is anticipated that approximately 30 participants will be recruited. Interviews will be audio recorded, transcribed verbatim and uploaded to NVivo V.11. The constant comparative method will be used to analyse the data, in order to produce a theoretical framework for mental health recovery that is grounded in the experiences of survivors.

**Ethics and dissemination** Ethical approval has been obtained from the Faculty of Medicine and Health Sciences Ethics Committee at the University of Nottingham. The findings of the study will be disseminated to academic, professional and survivor-based audiences to inform future policy developments and the provision of mental health recovery support to this population.

## Strengths and limitations of this study

► The study has been codeveloped with survivors of modern slavery.
► A lived experience advisory board will facilitate the involvement of survivor voices throughout the research process.
► Survivors who do not speak English will still have the opportunity to take part in the study, through the use of interpreters.
► Modern slavery survivors are not a homogenous population and the recruitment strategy may limit the relevance of the findings to particular subgroups.
► The theoretical framework will be developed based on a UK sample. This may limit applicability to other contexts.

is problematic. Globally, the number of vulnerable migrants at risk of trafficking is increasing, with an estimated 65 million people forcibly displaced globally in 2015.[4]

In the UK, research conducted by the Home Office in 2014[5] estimated that there were between 10 000 and 13 000 victims of modern slavery in the UK. However, the Home Office in 2017 has argued that this is the 'tip of the iceberg'.[6] Due to fear of arrest, deportation or ill treatment by the police, many survivors choose not to access the support provided through official channels, for example, the National Referral Mechanism (NRM) in the UK.[7] Therefore, official figures do not take into account the unknown numbers of people who are not reported or who continue to live in slavery. Despite these limitations, official statistics do highlight that more people are being freed and are therefore in need of support. The NRM 2019 End of Year Summary reports a 52% increase in referrals compared with 2018.[8] Official statistics also demonstrate that the survivor population within UK is diverse. Survivors originate from 130 different countries and the three

## INTRODUCTION

Slavery and human trafficking are crimes involving the violation of human rights and refer to exploitative situations where an individual cannot refuse or leave due to threats, coercion or abuse of power.[1,2] Activities involving slavery include forced labour exploitation, forced sexual exploitation, forced marriage and servitude. In 2016, it was estimated that approximately 40.3 million people are currently in slavery worldwide.[3] Predicting the future prevalence of slavery

most common nationalities are British, Albanian and Vietnamese.[8] In 2019, just over half of NRM referrals were exploited as adults, whereas 43% were minors.

Modern slavery (in the UK this term encompasses both slavery and human trafficking) presents a significant public health concern and disproportionately affects vulnerable individuals, such as young people, migrants and those living in poverty.[9 10] Survivors have often experienced extreme violence and psychological abuse.[11] Addressing the mental health needs of this population is therefore now part of antitrafficking policies in the UK such as the Modern Slavery Act,[12] as well as internationally, for example, in the Palmero Protocols.[13] Yet, the provision of evidence-based mental health support for this population is one of the largest gaps in both the national and global antislavery response.[14]

The literature that is available has predominantly focused on the mental health needs of women, including those trafficked for sexual exploitation in South or South East Asia and to a lesser extent Europe.[15] Abas and colleagues[16] identified that for women who returned home from the UK to Moldova after being sex trafficked, 55% continued to meet the diagnostic criteria for depression and Post Traumatic Stress Disorder (PTSD) 6 months after their return. Data from high-income countries (including the UK), for men and those who have experienced forced labour, are sparse.[15]

Despite the emerging evidence base demonstrating high levels of mental distress, tailored support is often found to be insufficient and inadequate leading to high levels of unmet need.[17] When survivors do make contact with mental health services, this is, frequently via adverse routes, for example, detention under the Mental Health Act 1983/2007. Within healthcare more generally, it is estimated that one in eight members of National Health Service (NHS) staff have had contact with survivors.[18] However, healthcare providers report that they feel unprepared to manage and support survivors. For example, in the Provider Responses, Treatment and Care for Trafficked People (PROTECT) studies[18] NHS staff reported not knowing what questions to ask people who may have experience modern slavery and 78% believed they had insufficient training to assist those who had been trafficked.

The limited evidence available from the UK highlights a complex picture in relation to the mental health needs of survivors. High prevalence rates of depression, anxiety, PTSD and suicidal ideation have been identified,[7] as well as unmet needs in relation to hearing voices.[7] For young people who have been trafficked, Attention Deficit Hyperactivity Disorder (ADHD) and adjustment disorders are common.[19] While these studies have highlighted the level of mental health need within the population, what is not clear is how individuals perceive/experience mental health; what resources they have to overcome their adverse experiences and what support or interventions would be helpful to facilitate post-traumatic growth (PTG) and recovery.[14]

PTG is defined as the individuals' psychological development after an adverse incident.[20] While traditional theories of resilience emphasise the importance of a return to normal after a traumatic event,[21 22] PTG recognises that adversity can lead to significant changes, some of which may be experienced as positive changes.[20] Rather than a problem or illness orientated position, PTG emphasises the active role of the individual in processing the event and constructing a new identity in the aftermath of trauma.[23] More recent work has started to apply the theory of PTG to clinical populations, for example, those with psychosis.[24] Through this personal growth process, the strengths and resources of the individual are emphasised, and positive adaptions learnt.

Within this context, modern slavery survivors are seen as active participants in their care, with resources and strengths that can be used for mental health recovery.[25] This can include factors within the individual, such as coping strategies, and also those external to them, for example, social networks. Recovery in this context is therefore a subjective experience and has been defined as 'a deeply personal, unique process of changing one's attitudes, values, feelings, goals, skills and/or roles' and as 'a way of living a satisfying, hopeful and contributing life even within the limitations caused by illness'.[26]

Recovery has become the underpinning discourse for mental health service provision globally[26] and in England.[27] A key insight into this perspective is that living well involves less emphasis on symptom amelioration, and a stronger focus on addressing psychological and social needs, supporting self-management and building individual and community resilience.[28] This understanding is relevant to work with survivors of modern slavery, who may have mental health symptoms (eg, those related to trauma) and also a range of other psychological (eg, self-identity), social (eg, anticipated and experienced discrimination) and cultural challenges (eg, dislocation).

The aim of this study is to develop a theoretically informed understanding of mental health recovery for survivors of modern slavery within the UK. The objectives are (1) to understand the mental health needs and strengths of modern slavery survivors, (2) to explore with modern slavery survivors the concept of mental health recovery; what it means to them and how it could be promoted and (3) to construct a theoretically informed framework for mental health recovery, based on the experiences of modern slavery survivors, which can inform a future intervention.

## METHODS AND ANALYSIS
### Design
A qualitative study informed by grounded theory methodology based on individual, semistructured interviews with modern slavery survivors is proposed. Widely used in health and social sciences, grounded theory is an inductive methodology that facilitates the development of a

theoretical framework grounded in the experiences of participants.[27]

## Setting

Participants will be recruited from relevant community-based non-governmental organisations (NGOs). This will include both NRM and non-NRM provider organisations.

## Sample

Eligibility criteria:

1. Self-identify or be identified by an NGO as a survivor of modern slavery. Self-identifying involves describing being in situations or having experiences that align with modern slavery, even if the term 'modern slavery' is not used by the participant.
2. Be currently in the UK.
3. Be aged over 18 years.
4. Have exited slavery for more than 12 months.
5. Be able to participate in an interview process conducted in English, either with or without the assistance of an interpreter.
6. Be able and willing to discuss issues related to mental health/distress. A formal diagnosis of a mental health problem is not required, although potential participants should have one of the following (a) self-reported clinical diagnosis, (b) recent (in the last 2 years) primary or secondary mental health service use or (c) self-reported recent (in the last 2 years) experience of mental distress

## Procedures

Aligned with the study design, theoretical sampling will be used. This is an iterative process whereby data collection and analysis occur simultaneously, allowing insights into preliminary analysis to inform the recruitment of future participants. This will occur until data saturation is reached, that is, no new insights or themes are identified.[29] After an initial sample of participants have been recruited and their data analysed, future sampling will address potentially any of the following:

1. Gaps in relation to participant demographics (eg, gender, age, ethnicity, experience of a particular form of slavery).
2. Emergent theoretical concepts, for example, to understand or elucidate a particular insight into more detail.
3. Any explanatory gaps between the concepts identified, for example, to demonstrate the relationship between two different concepts.

If necessary, theoretical sampling will be supplemented with snowball sampling techniques, whereby participants are asked to identify from within their networks other individuals to approach for interview. It is acknowledged that snowball sampling could potentially narrow the demographic focus of participants and so will only be used once other recruitment methods have been exhausted. The nature of theoretical sampling and achieving data saturation means that determining in advance the precise sample size is problematic. Previous grounded theory studies suggest that 30 interviews are likely to be sufficient to achieve saturation.[30 31] Therefore, this study will recruit, at least, 30 modern slavery survivors or until data saturation is achieved.

In-depth, semistructured interviews will be conducted with participants. Distress protocols have been put in place in order to protect the research participants, the researcher undertaking the interviews, interpreters and transcribers. The researcher will receive relevant training to ensure that they have the skills required. They will also receive monthly clinical supervision to assist with the processing of any difficult material or interviews. Participants will be made aware that if they make disclosures which indicate they are at risk this may need to be escalated to the appropriate authorities. Signposting materials to support organisations will also be available to participants should these be required.

Where necessary, independent NHS-approved interpreters will be used during the interviews and written materials translated. The interview schedules were developed in collaboration with the Lived Experience Research Advisory Board and informed by the World Health Organisation (WHO) recommendations for interviewing trafficked women.[32] As per the iterative nature of grounded theory, the interview schedules will continue to evolve during the course of the project to address emergent concepts and ideas with participants.

Recruitment of participants will take place via relevant NGOs who provide support to survivors of modern slavery, and by advertising the study widely through social media. Twitter will be the main platform used for recruitment. The recruitment poster will be shared and direct messaging to the researcher and principal investigator made possible in the event of any questions. The research team will work with the NGOs involved to distribute study information and identify people who meet the eligibility criteria. Individuals who are interested in participating in the study will be asked to contact the research team by either phone or email. After explaining the study purpose (face to face, by phone or online depending on the participant's preference), including the voluntary and anonymous nature of participation and distributing the information sheet, informed written consent will be obtained. All written documentation will be provided to participants in the language of their choice. The interviews (approximate duration 60 min) will be audio-recorded and transcribed verbatim. They will be conducted at a time of the participants choosing, whenever possible as a face-to-face interview in a place which assures the participant of safety, privacy and comfort. Where this is not possible, interviews will be offered via telephone or video call and the participant will be able to choose based on personal preference which they would like. Audio recordings using an encrypted device will be made for these interviews too.

## Data analysis

Interview transcripts will be uploaded to NVivo V.11 software for analysis. The constant comparative method will be used to generate successively more abstract concepts and theories through the inductive coding of individual transcripts.[33] The codes identified will then be compared against those generated in other transcripts. This will identify both consistency and any divergence in the theoretical framework being constructed. Finally, the emergent theoretical framework will be compared against existing models and theories of recovery in mental health, such as connectedness, hope, identity, meaning and empowerment.[34] They will assist with the identification of both what is novel for the survivor population and also what is similar and adapted from other populations.

## Patient and public involvement

Patient and public involvement is a core component of the project. As per guidelines,[35] survivors are engaged in the five key stages of the research process:
1. Design of the research.
2. Development of the grant application/preprotocol work.
3. Undertaking or managing the research.
4. Analysis of data.
5. Dissemination of findings.

A survivor led, lived experience research advisory board (LE -RAB) has been developed in collaboration with the Survivor Alliance (www.survivoralliance.org), a non-profit organisation that unites and empowers survivors of slavery and human trafficking worldwide. The role of the LE-RAB will be to provide a survivor perspective on all aspects of the study, including language and format of consent forms and information sheets, recruitment strategy, data analysis and dissemination. The group will meet six times during the course of the project.

## ETHICS AND DISSEMINATION

Ethical approval for the study was obtained from the Faculty of Medicine and Health Sciences, University of Nottingham in December 2019 (reference number 436–1912). The main ethical concern of the project is the potential to cause emotional or psychological distress to participants, if they recount their traumatic experiences. The project aims to minimise this by not specifically asking questions relating to the original trauma or trafficking experience. Instead, the interview will focus on mental health recovery and what this means to individuals. It is acknowledged that individuals may choose to recount their difficult experiences and if they do so, the distress protocol (online supplemental information 1) will be adhered to. All the questions in the interview schedule have been reviewed and approved by the lived experience LE-RAB who felt there was an important distinction to be made between survivors choosing to bring up topics themselves as opposed to being asked to talk about their trafficking experiences. Initial drafts of questions were amended according to the advice of LE-RAB members. They did not feel that the final questions included in the interview schedule would be triggering for other survivors.

Members of the LE-RAB also felt it is important to explicitly note in the interview schedule that mental health support is not available as part of survivor's participation in the study; the researcher is not able to provide legal or mental health support. However, individuals will be made aware that they can contact the researcher to find out about mental health services in their local area. Due to the study being nationwide, this information cannot be included on the debriefing information sheet. All participants will receive details of national support available (eg, Samaritans helpline).

Participants will remain anonymous, their names (and any other identifiable details) will be replaced by a pseudonym at the point of transcription. Individuals' identities will not be disclosed in the final report or any subsequent publications. Secure electronic and paper-based filing systems for both the recordings and transcriptions will be established. All study related that data will be kept for a minimum of 7 years after the study end date.

The findings of the research will be disseminated to academic, professional, survivor and policy audiences via written publication, verbal presentations and knowledge-sharing events.

**Contributors** NW and MD conceived of the research idea and with KB, CB, MJ and MS designed the study and secured funding. EH wrote the initial draft of this paper. LERAB contributed to the development of the research materials. All authors have read, edited and revised the paper.

**Funding** This study is funded by the National Institute for Health Research (NIHR) (Research for Patient Benefit, project reference PB-PG-1217–20036). Mike Slade acknowledges the support of Centre for Mental Health and Substance Abuse, University of South-Eastern Norway and the NIHR Nottingham Biomedical Research Centre. The views expressed are those of the authors and not necessarily those of the NIHR or the Department of Health and Social Care.

**Competing interests** None declared.

**Patient consent for publication** Not required.

**Ethics approval** The study was approve by the Faculty of Medicine and Health Sciences Ethics Committee at the University of Nottingham.

**Provenance and peer review** Not commissioned; externally peer reviewed.

**ORCID iD**
Emina Hadziosmanovic http://orcid.org/0000-0001-5154-0136

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
