## [Reviewer comments · BMJ Open]

ARTICLE DETAILS

TITLE (PROVISIONAL)	Mental Health Recovery for Survivors of Modern Slavery: Grounded Theory Study Protocol
AUTHORS	Wright, Nicola; Hadziosmanovic, Emina; Dang, Minh; Bales, Kevin; Brookes, Caroline; Jordan, Melanie; Slade, Mike; Research Advisory Board, Lived Experience

VERSION 1 – REVIEW

REVIEWER	Elizabeth Such SchARR, University of Sheffield UK
REVIEW RETURNED	15-Apr-2020

GENERAL COMMENTS	This is a well constructed protocol for a qualitative study of the meaning of mental health recovery for people who have been victims of modern slavery. There are some minor concerns relating to some aspects of the protocol highlighted in the text. I think there needs to be greater consideration of the potential diversity within the sample and the challenges the researchers face addressing diversity in the sample population. Translating these concerns into a framework that is useful and operationalisable may be a significant challenge. Overall, an important piece of work. The reviewer provided a marked copy with additional comments. Please contact the publisher for full details.
---

REVIEWER	Elizabeth Metcalf Cardiff University
REVIEW RETURNED	24-Apr-2020

GENERAL COMMENTS	The proposed study explores a very important area of care where there is little existing evidence and I hope the findings will inform future improved healthcare experience for these individuals. A key strength is that you are ensuring survivors voice is heard not only in the results, but informing the research methodology as well. It would be helpful to include an overview of the challenges of delivering care to this group of individuals in both primary and secondary care, to help contextualise how this study will inform future care. I think it would be helpful to specify this is in the context of individuals who currently remain within the UK, as I suspect the experiences of patients identified who subsequently leave the UK would be different, and this would impact the utility of outcomes globally.
--

	Regarding choice of interpreters where needed, you need to specify using only an appropriately trained interpreter as this will be a conversation requiring much skill and sensitivity to avoid risk of causing harm. Regarding participant selection, you will need to be careful to ensure representative demographics, including those subjected to modern slavery of UK origin. This will be essential in order for the overall project findings to meaningfully inform future delivery of care. Similarly, I would question your use of snowball sampling. I think this will risk findings becoming of narrow demographic focus so needs to be considered iteratively as ongoing recruitment takes place, widening perspectives from broader sampling. As mentioned above, I think the interviewers and interpreters will have to have some very specific training in terms of how they should manage information that may be gathered from these interviews- legal frameworks and escalation of concern/ safety netting. How will you follow up participants and manage any newly revealed clinical needs and legal issues? Consent and patient information literature- will be required in all languages etc of those participating, some planning required to achieve this but this shouldn't be insurmountable. Regarding telephone interviewing, I'd be uncomfortable about telephone interviews and would want assurance that appropriate safety netting is in place to support participants and assure they are safe if not physically with the interviewer. Might also be worth considering ensuring a third party is present for each interview to support process, though apologies I'm not sure whether there is precedent for this? Your third party might be a member of the survivor led advisory group. I think that is almost inevitable that these individuals will have significant mental and other health conditions- we already know this from existing research evidence. Therefore you will need robust measures to be put in place to ensure the ongoing safety and support for these vulnerable individuals. I don't think enough to say the subject won't be discussed as mental recovery is surely inherently linked? More detail here is definitely needed to ensure no harm comes to participants as a result of the study. Overall a good proposal, but some revision is needed to ensure the most useful and applicable data is obtained to inform your answering of the research questions, and improved participant safety netting is needed. The reviewer provided a marked copy with additional comments. Please contact the publisher for full details.
--	--

VERSION 1 – AUTHOR RESPONSE

Reviewer Elizabeth Such Comments (changes highlighted in green on text):	Author Responses
Clearer definition of both modern slavery and	On page 2 in the abstract we have added the following text: Slavery and human trafficking are

human trafficking	crimes involving the violation of human rights and refer to exploitative situations where an individual cannot refuse or leave due to threats, coercion or abuse of power. Activities involving slavery include forced labour exploitation, forced sexual exploitation, forced marriage and servitude.
In addition, I assume the sample will include people with very different social and cultural backgrounds, including domestically trafficked persons. Is it possible to develop a cross-cultural theoretical framework in this context	The following points have been added to the strengths and limitations section:  • Modern slavery survivors are not a homogenous population and the recruitment strategy may limit the relevance of the findings to particular sub-groups. • The theoretical framework will be developed based on a UK sample. This may limit applicability to other contexts.
This is an estimate at a given time period (2016?)	The time period 2016 has been added to the text on page 3.
A note on prevalence by nationality may be helpful - NB largest proportion of NRM referrals are UK victims. Also, minors vs. adults	The following text has been added to page 3: The NRM 2019 End of Year Summary reports a 52% increase in referrals compared to 2018(8). Official statistics also demonstrate that the survivor population within UK is diverse. Survivors originate from 130 different countries and the three most common nationalities are British, Albanian and Vietnamese(8). In 2019, just over half of NRM referrals were exploited as adults whereas 43% were minors.
Also ref: Such, E, C Laurent, R Jaipaul, and S Salway. 2020. "Modern Slavery and Public Health: A Rapid Evidence Assessment and an Emergent Public Health Approach." Public Health 180: 168–79.	This reference has been added to the text and the reference list.
Studies indicative only of prevalence of mental health morbidity. Is it also important to look at how people experience mental health?	The following has been added to the text on page 4: Whilst these studies have highlighted the level of mental health need within the population, what is not clear is how individuals perceive/experience mental health.
Not sure what mental health 'growth' means	The use of the word growth has been amended in the text on page 4 to Post-Traumatic Growth for consistency. A definition of this term is included on page 4: Post Traumatic Growth (PTG) is defined as the individuals' psychological development after an adverse

	incident (20).
Define resilience - multiple definitions, some social so needs clarifying.	The following references have been added to clarify the definition of resilience that is being used in this context: Higgins, GO. Resilient Adults: Overcoming a Cruel Past. Jossey-Boss: San Francisco. 1994. Wollin, SJ. and Wollin, S. The Resilient Self: How Survivors of Troubled Families Rise Above Adversity. Villard Books: New York. 1993.
Not sure this last sentence is needed or necessarily follows. Remove?	This sentence has now been removed from the text on page 4: It ultimately means that individuals are supported to live as well as possible.
Can this approach be truly 'grounded' if you have the a priori position above? Moreover, an approach informed by grounded theory ...?	This has been clarified on page 5 by the addition of the following: A qualitative study informed by grounded theory methodology based on individual, semi-structured interviews with modern slavery survivors is proposed.
So these settings are community based? People receiving community support? NRM support? Clarify the 'settings'	The following has been added to the main text on page 5: Participants will be recruited from relevant community based Non-Governmental Organisations (NGOs). This will include both NRM and non-NRM provider organisations.
What constitutes 'recent'?	This has been clarified on page 5: (b) recent (in the last two years) primary or secondary mental health service use; OR (c) self-reported recent (in the last two years) experience of mental distress
Contradiction - say problematic then identify a sample size. 'Around' or 'At least' 30 ?	The words "at least" have been added as suggested by the reviewer.
Can you say more about this? Very broad - Instagram, twitter, using DMs?	The following has been added to page 6: Twitter will be the main platform used for recruitment. The recruitment poster will be shared and Direct Messaging to the researcher and principal investigator made possible in the event of any

	questions.
Is the protocol conducted face-to-face? Clarify	The following has been added to page 6 to clarify this point: After explaining the study purpose (face to face, by phone or online depending on the participants preference), including the voluntary and anonymous nature of participation, and distributing the information sheet, informed written consent will be obtained.
Consider more strengths/limitations. Cultural equivalence, exclusion of non-English speakers, generalisability of framework. Can you comment on how existing models of recovery are being used to inform your analysis?	Following the editors advice we have removed the strengths and limitations section from the main text. As outlined above acknowledgement is given to the potential transferability issues. Please note that non-English speakers will be included within the sample. With regards to the comment relating to existing models of recovery, the following has been added to data analysis section on page 7: Finally, the emergent theoretical framework will be compared against existing models and theories of recovery in mental health, such as CHIME (connectedness, hope, identity, meaning and empowerment) (35). The will assist with the identification of both what is novel for the survivor population and also what is similar and adapted from other populations.
Reviewer Elizabeth Metcalf Comments (changes highlighted in yellow on text):	Author Responses:
Strength – ensuring survivors voice in not only results but in research methodology	Many thanks for this positive comment with regards to our approach to working with survivors.
Worth also briefly discussing the challenges of delivering care to this group of individuals in both primary and secondary care	To address this point the following paragraph has been added on page 3 and 4: Despite the emerging evidence base demonstrating high levels of mental distress, tailored support is often found to be insufficient and inadequate leading to high levels of unmet need (17). When survivors do make contact with mental health services this is frequently via adverse routes, for example detention under the Mental Health Act 1983/2007. Within health care more generally, it is estimated that one in eight members of NHS staff have had contact with survivors (18). However, health care providers report that they feel unprepared to manage and support survivors. For example in

	the PROTECT studies(18) NHS staff reported not knowing what questions to ask people who may have experience modern slavery and 78% believed they had insufficient training to assist trafficked people.
I think it would be helpful to specify this is in the context of individuals who currently remain within the UK. I suspect the experiences of patients identified who subsequently leave the UK would be different.	The point has been clarified within the project aims on page 4: The aim of this study is to develop a theoretically informed understanding of mental health recovery for survivors of modern slavery within the UK.
Specifically an appropriately trained interpreter as this will be a conversation requiring much skill and sensitivity to avoid risk of causing harm	NHS approved interpreters will be used within the study. This is more clearly stated within the text on page 6: Where necessary, independent NHS-approved interpreters will be used during the interviews and written materials translated.
Essential in order for the overall project findings to meaningfully inform future delivery of care	Thank you for this comment. The authors agree. No amendments required on the text.
I think this will risk findings becoming of narrow demographic focus so needs to be considered iteratively as ongoing recruitment takes place, widening perspectives from broader sampling	Snowball sampling will only be used once all other methods have been exhausted. The following text has been added to page 6 to acknowledge the potential challenges with this approach: It is acknowledged that snowball sampling could potentially narrow the demographic focus of participants and so will only be used once other recruitment methods have been exhausted.
I think the interviewers and interpreters will have to have some very specific training in terms of how they should manage information that may be gathered from these interviews – legal frameworks and escalation of concern/safety netting. How will you follow-up participants and manage any newly revealed clinical needs and legal issues?	Distress protocols have been developed which specify the steps to be taken. These are available as supplementary resources. To clarify this, the following has been added to page 6: Distress protocols have been put in place in order to protect the research participants, the researcher undertaking the interviews, interpreters and transcribers. The researcher will receive relevant training to ensure they have the skills required. They will also receive monthly clinical supervision to assist with the processing of any difficult material or interviews. Participants will be made aware that if they make disclosures which indicate they are at risk this may need to be escalated to the appropriate authorities. Signposting materials to support organisations will also be available to participants should these be required. In addition to the information on page 6, the following has also been added to the ethics

	section on page 8: Members of RAB also felt it important to explicitly note in the interview schedule that mental health support is not available as part of survivor's participation in the study; the researcher is not able to provide legal or mental health support. However, individuals will be made aware that they can contact the researcher to find out about mental health services in their local area. Due to the study being nation-wide, this information cannot be included on the debriefing information sheet. All participants will receive details of national support available (e.g., Samaritans helpline).
Required in all languages etc. of those participating, some planning required but no insurmountable.	All study documents will be made available in the participants' language of choice. This is stated within the main text.
I'd be uncomfortable about telephone interviews and would want assurance that appropriate safety netting is in place to support participants and assure they are safe if not physically with the interviewer. Might also be worth considering ensuring a third party is present to support process, though apologies I'm not sure whether there is precedent for this?	To ensure participants have as much control over the interview process as possible, they will be offered different options. For example some survivors may prefer the added anonymity that a telephone interview offers. Similarly, for those who are residing in safe house accommodation face to face interviews are not an option as the participants will not be able to reveal their address to the research team. The option of having a third party present has been considered by the lived experience expert on the research team and it was felt that was not a feasible option.
Perhaps a member of this group could be the 3rd party support in interviews?	Please see previous comment regarding this.
I think this is almost inevitable and therefore will need robust measures to be put in place to ensure the ongoing safety and support for these vulnerable individuals. I don't think enough to say the subject won't discussed as mental health recovery is surely inherently linked	The following text has been added to the text on page 7: All the questions in the interview schedule have been reviewed and approved by the Lived Experience Research Advisory Board (RAB) who felt there was an important distinction to be made between survivors choosing to bring up topics themselves as opposed to being asked to talk about their trafficking experiences. Initial drafts of questions were amended according to the advice of RAB members. They did not feel that the final questions included in the interview schedule would be triggering for other survivors.